# SARS-CoV-2 rapidly evolves lineage-specific phenotypic differences when passaged repeatedly in immune-naïve mice
Julian Daniel Sunday Willett [1,2,3], Annie Gravel[4], Isabelle Dubuc[4], Leslie Gudimard[4], Ana Claudia dos Santos Pereira Andrade[4], Émile Lacasse [4], Paul Fortin [4,5,6], Ju-Ling Liu[2,7], Jose Avila Cervantes [2,7], Jose Hector Galvez [8], Haig Hugo Vrej Djambazian [2,7], Melissa Zwaig[2,7], Anne-Marie Roy[2,7], Sally Lee[2,7], Shu-Huang Chen[2,7], Jiannis Ragoussis [2,7] ✉ & Louis Flamand [4,9] ✉

The persistence of SARS-CoV-2 despite the development of vaccines and a degree of herd immunity is partly due to viral evolution reducing vaccine and treatment efficacy. Serial infections of wild-type (WT) SARS-CoV-2 in Balb/c mice yield mouse-adapted strains with greater infectivity and mortality. We investigate if passaging unmodified B.1.351 (Beta) and B.1.617.2 (Delta) 20 times in K18-ACE2 mice, expressing the human ACE2 receptor, in a BSL-3 laboratory without selective pressures, drives human health-relevant evolution and if evolution is lineage-dependent. Late-passage virus causes more severe disease, at organism and lung tissue scales, with late-passage Delta demonstrating antibody resistance and interferon suppression. This resistance co-occurs with a de novo spike S371F mutation, linked with both traits. S371F, an Omicron-characteristic mutation, is co-inherited at times with spike E1182G per Nanopore sequencing, existing in different within-sample viral variants at others. Both S371F and E1182G are linked to mammalian GOLGA7 and ZDHHC5 interactions, which mediate viral-cell entry and antiviral response. This study demonstrates SARS-CoV-2's tendency to evolve with phenotypic consequences, its evolution varying by lineage, and suggests non-dominant quasi-species contribution.

Part of the difficulty in responding to the COVID-19 pandemic has been predicting viral evolution and its impact on clinically relevant traits, such as disease severity, infectivity, and treatment resistance[1]. The basic biology of the severe acute respiratory syndrome Coronavirus 2 (SARS-CoV-2), the virus that causes COVID-19, is well understood[2]. However, knowledge of SARS-CoV-2 evolution is limited. Many RNA viruses, like SARS-CoV-2, exist as quasispecies, meaning viral populations contain a multitude of mutants subjected to continuous selection, competition, and genetic variation[3–9]. These alleles give plasticity to the viral population allowing for

rapid adaptation/selection to a variety of circumstances[10]. Viral evolution is also complex and affected by several factors, including host genetic background[11], host immune status[12], the organs targeted by the virus[13], and a population's collective immunity[14]. As a result, it has been difficult to predict the emergence of new variants with altered virulence.

Several SARS-CoV-2 variants of concern (VOC) have emerged, each with mutations providing dissemination advantages[1]. The SARS-CoV-2 Alpha variant (B.1.1.7) gained a growth advantage and rapidly spread globally due to the spike protein N501Y mutation that enhanced affinity for

[1]Quantitative Life Sciences Ph.D. Program, McGill University, Montreal, QC, Canada. [2]McGill Genome Centre, McGill University, Montreal, QC, Canada. [3]Lady Davis Institute, Jewish General Hospital, Montreal, QC, Canada. [4]Axe maladies infectieuses et immunitaires, Centre de Recherche du Centre Hospitalier Universitaire de Québec- Université Laval, Québec, Canada. [5]Centre de Recherche ARThrite-Arthrite, Recherche et Traitements, Université Laval, Québec, QC, Canada. [6]Division of Rheumatology, Department of Medicine, CHU de Québec-Université Laval, Québec, QC, Canada. [7]Department of Human Genetics, McGill University, Montreal, QC, Canada. [8]Canadian Centre for Computational Genomics, McGill University, Montreal, QC, Canada. [9]Département de microbiologie-infectiologie et d'immunologie, Université Laval, Québec, QC, Canada. ✉e-mail: ioannis.ragoussis@mcgill.ca; louis.flamand@crchudequebec.ulaval.ca

the cellular entry receptor, angiotensin-converting enzyme 2 (ACE2)[15]. Other VOC, such as the Delta (B.1.617.2) variant, gained advantages with additional spike substitutions, including L452R[16], T478K[17], and P681R[18] that affect viral transmissibility and antibody neutralization for naturally or artificially vaccinated individuals[19]. To better predict changes in SARS-CoV-2 and reduce the impact of a future pandemic, time-effective models must be defined for studying viral changes over successive generations and characterizing their evolutionary consequences.

In 2020, Gu et al. studied short-term viral evolution by infecting aged mice with the SARS-CoV-2 reference strain (IME-BJ05) and collecting lung tissue at set timepoints following infection, using this isolate to infect successive generations of mice[20]. While wild-type mice are considered to be less susceptible to the virus since mouse Ace2 is not used by SARS-CoV-2 as its receptor, Gu et al. demonstrated that serial passaging of infected lung homogenate across mice produced a mouse-adapted strain (MASCp6) that causes pulmonary symptoms, matching adaptations observed by others who previously applied the technique to influenza[20,21]. After six passages, they found several clinically relevant genetic changes, including the de novo appearance of A23063T that causes the spike mutation N501Y[20]. This mutation is associated with enhanced viral infection and transmission[15]. Subsequently, the group passaged the MASCp6 isolate an additional 30 times and characterized the MASCp36 isolate[22]. MASCp36 contained several mutations (K417N, Q493H, N501Y) in the spike protein conferring greater mouse Ace2 receptor binding affinity, increased infectivity and greater virulence[22]. Others replicated this study using the same viral variant, observing similar development of clinically relevant variant alleles[23].

Since these studies, new VOC have arisen with the latest being Omicron, negatively impacting treatment efficacy[1]. Others have studied viral dynamics using recent lineages in serial passages in cells, observing similar development of directional selection and clinically concerning mutations with contributions by quasi-species[24,25]. Understanding how SARS-CoV-2 evolves as it passes between mammalian hosts is pressing. We adapted Gu et al.'s approach[20] to determine if more recent and human-adapted lineages, B.1.351 (Beta) and B.1.617.2 (Delta), evolved over a twenty passage study in transgenic mice expressing the human ACE2 receptor, if this evolution varied by lineage, and whether these changes were relevant to human health. We used transgenic animals to limit selective pressure and adaptations to the murine ACE2 receptor, as occurred in Gu et al.'s works[20,22]. While others have studied COVID-19 in these mice[26–28], we did not locate papers that emphasized viral evolution. Here we report on virus evolving in the setting of no experimental selective pressures with changes observed at the organism, tissue, and genetic scale, this evolution varying by lineage with solely late-study Delta lineage presenting characteristics resulting in decreased antibody neutralization, with variant alleles arising de novo that have been linked with these phenotypes.

## Results

### Late passaged virus produced more viral RNA, greater weight loss, and worse lung inflammation. Only late-study Delta was associated with increased viral load and cytokine changes linked to severe COVID-19

To study evolution on the organism level, we compared clinical scores of mice infected with virus before we passaged it in mice (passage [P] 0, or early-passaged virus) and after twenty passages (P20, or late-passaged virus), following a protocol adapted from the work of Gu et al. (Fig. 1)[20,22]. Unlike Gu et al.'s study that required adaptation of SARS-CoV-2 to a murine Ace2 receptor for infectivity, our study used transgenic mice expressing the human ACE2 receptor in airway, liver, kidney, and gastrointestinal epithelium, making the results more relevant for humans[20]. On day 3 post-infection, lungs of passage [P]20 Beta and Delta infected mice contained significantly more SARS-CoV-2 RNA than corresponding P0 viruses ($p < 0.05$ and $p < 0.008$ for Beta and Delta lineage, respectively) (Fig. 2a). Mice infected with P20 Delta virus had a significant increase (6.3 x) in lung viral loads relative to P0 (Fig. 2b) ($p = 0.008$) with no significant difference for Beta P0 and P20 (Fig. 2b). Both P20 Beta and Delta viruses caused

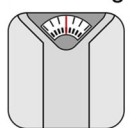
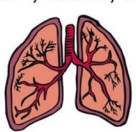
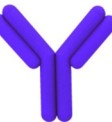
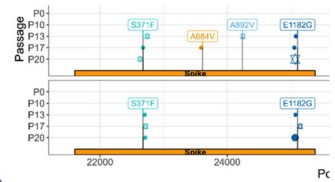

**Fig. 1 | Study design.** SARS-CoV-2 ACE2-adapted mice were inoculated with either B.1.351 (Beta) or B.1.617.2 (Delta) virus and left for 3 days for virus to proliferate. They were then sacrificed with the lung homogenate used to infect another mouse. This was repeated for ten passages. P10 virus for each lineage was then used to infect three additional mice and the process was repeated for an additional ten passages. P0 and P20 virus for each lineage was subsequently used to infect additional mice to compare weight loss and survival between early and late passage virus, and test non-adapted mouse susceptibility to the virus. Antibody neutralization of virus was also compared.

significantly more rapid and greater weight loss than P0 3 days after infection (10% body weight for Beta, 5% body weight for Delta, $p < 0.001$ for both tests) with similar survival curves, with less significant differences by passage for Delta virus (Fig. 2c, d).

To determine if the significant increase in lung viral load observed by Delta virus was related to growth kinetic differences between P0 and P20 viruses, we performed in vitro growth curve analysis. Using two independently evolved P20 Delta viruses, we observed that P20 viruses yielded significantly (4-5X) more infectious virus after 24 h of infection than Delta P0 virus ($p = 0.02$) (Fig. 3). This difference in viral titers was no longer observed after 48 h of infection (Fig. 3).

Lungs of mice infected with P0 and P20 viruses were examined for signs of inflammation as assessed by histology and presence of inflammatory cytokines[29]. On day 6 post-infection, P20 Beta and Delta viruses caused greater leukocyte infiltration, red blood cell extravasation, and decreased alveolar space versus P0 and mock-infected mice (Fig. 4a–e). Lung tissues were also analyzed for SARS-CoV-2 antigenic burden and leukocyte recruitment as measured using anti-N and anti-CD45 antibodies, respectively[28]. Results indicated that significantly more cells were expressing N by Beta P20 than P0, with no change by passage for Delta (Fig. 5). While Delta had significantly greater immune cell infiltration than Beta per CD45 staining, immune infiltration did not significantly change between P0 and P20 for either lineage (Fig. 5). On day 3 post infection, Delta P20 infection resulted in diminished IFN-β ($p < 0.05$) and IFN-γ ($p < 0.05$) production versus P0, which co-occurred with increased viral load (Fig. 3) and has been

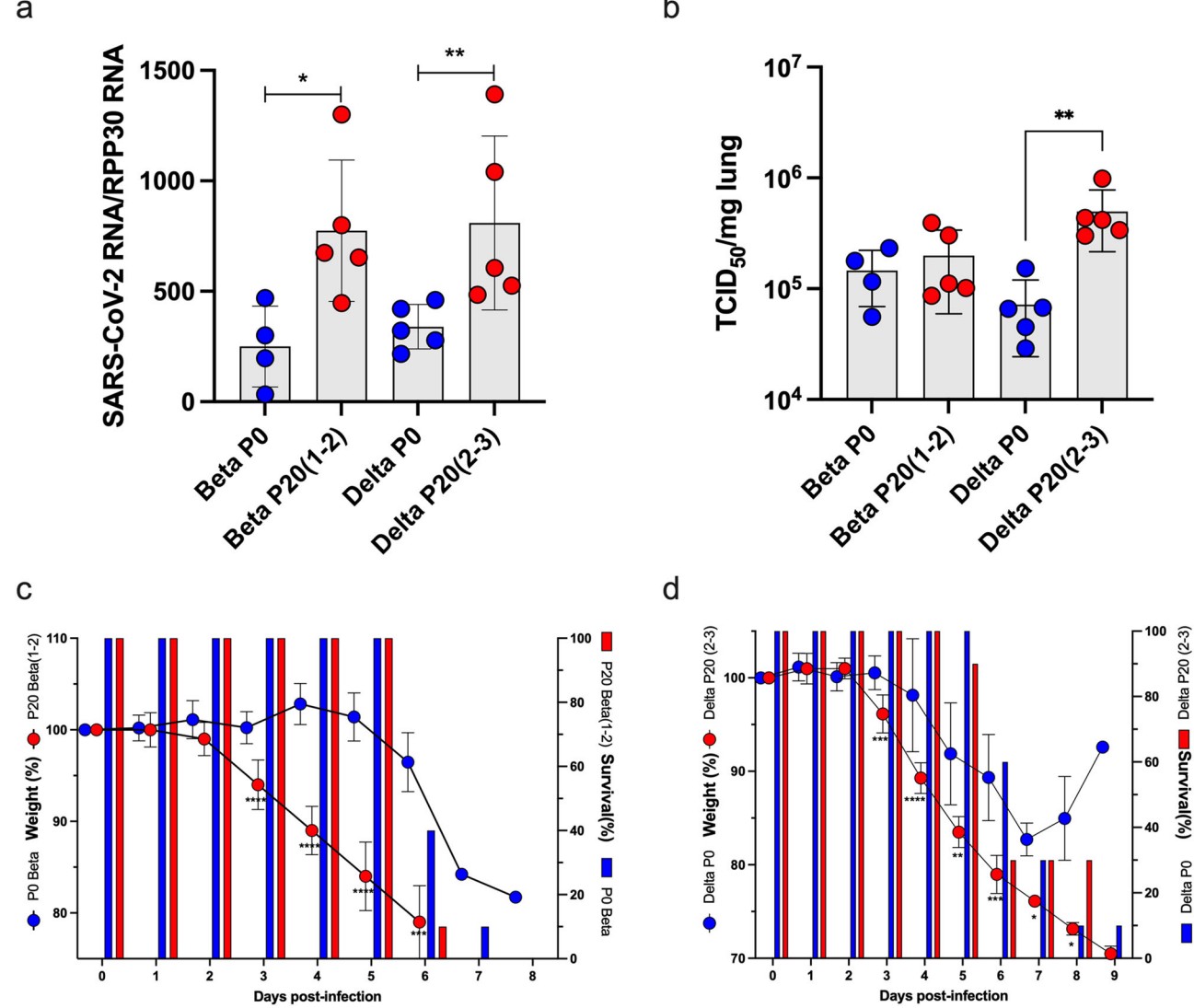

**Fig. 2 | Pathogenesis of P0 and P20 viruses in K18-ACE2 mice.** Mice ($n = 15/$ group) were infected with 500 $TCID_{50}$ of P0 or P20 (1-2) Beta and P20 (2-3) Delta viruses. **a, b** On day 3 post-infection, mice ($n = 5$) were euthanized and lungs collected for SARS-CoV-2 RNA (**a**) and infectious viral load (**b**) determination. Results are expressed as mean ± SD SARS-CoV-2 RNA copies/RPP30 RNA copies (**a**) and mean ± SD SARS-CoV-2 $TCID_{50}$/mg of lung tissue (**b**). *$p < 0.05$ and **$p < 0.008$ as determined using non-parametric two-sided Mann–Whitney test. **c, d** The weight and survival of each mouse ($n = 10$/group) was monitored daily throughout the experiment. Weights are reported as mean percentage ± SD of weight relative to day 0. The percent survival in each group is presented. *$p < 0.05$, **$p < 0.01$, ***$p < 0.001$, ****$p < 0.0001$ as determined using unpaired, two-tailed $t$-tests assuming no difference between groups.

linked with more severe COVID-19 in humans (Fig. 6a, b)[30,31]. Beta P20 virus induced greater production of proinflammatory CXCL1 ($p = 0.051$), CCL2 ($p < 0.005$) and IL-6 ($p < 0.05$) versus P0 (Fig. 6c–e)[32].

We next assessed whether repeated passages in K18-ACE2 mice would select for viruses capable of infecting non transgenic mice. C57BL6 (B6) mice were infected intranasally with P0 and P20 Beta and Delta viruses isolated from serially-infected K18-ACE2 mice. Lung SARS-CoV-2 RNA and infectious viral loads were determined 3 days later. Both P0 and P20 Beta viruses successfully replicated in B6 mouse lungs with both viruses producing similar SARS-CoV-2 RNA and infectious viral loads (Supplementary Fig. 1). None of the P0 or the P20 Delta virus infected mice had detectable SARS-CoV-2 RNA or infectious viral loads above the limit of detection (Supplementary Fig. 1).

### Late-study Delta, not Beta, virus presented with greater antibody resistance

We next determined the ability of sera from vaccinated human subjects to neutralize passaged viruses, comparing it to stock P0 viruses. Consistent with previous reports, sera from vaccinated subjects had varying neutralizing potential depending on the variant, with neutralization titers being greatest against the original WT strain and lowest for the Beta isolate (Fig. 7a)[33,34]. The same sera were analyzed for their ability to neutralize P20 viruses. P20 Beta was neutralized equally (13 out of 24) or more efficiently (8 of 24) that the P0 virus (Fig. 7b). In contrast, P20 Delta virus was significantly less sensitive to antibody neutralization by vaccinated subject sera than P0 ($p < 0.0002$) with 15 of 24 sera showing decreased neutralizing potential toward the P20 Delta virus relative to P0 (Fig. 7c).

### Variant alleles associated with COVID-19 severity and susceptibility, cytokine suppression, and antibody resistance arose de novo predominantly in late passaged Delta virus

To link phenotypic changes with genetics, we identified those variant alleles that changed in frequency across passages, including those that arose de novo and disappeared, and annotated them for traits with implications on public health, such as conferring antibody resistance[20,35]. Variant allele frequency across passages and between samples in single passages did not

statistically change (adjusted $p$-value $\geq 0.05$), except when comparing Beta P17 and P20 to earlier passages and Delta P10 and P20 (Supplementary Fig. 2a, b). Given no significant change in mean allele depth, these differences could be due to growing allelic diversity and within-host evolution of quasi-species, per Tonkin-Hill et al.[36]. Sample contamination was less likely as there was no significant difference in variant allele frequency (VAF) or depth between samples for either viral lineage (Supplementary Fig. 2c–d).

Of 82 unique variant alleles, there were 26 that were nonsynonymous and appeared de novo or whose VAF clearly changed with 20 having immediately clinically-relevant annotations, with 20 occurring in viral proteins that interact with human proteins, 1 believed to confer drug resistance, 1 associated with the Omicron lineage, 1 linked with antibody escape, and 7 at vaccine targets (Table 1, Supplementary Table 1)[35,37]. These included 9 that were specific to Beta, 16 that were specific to Delta, and 1 that changed in

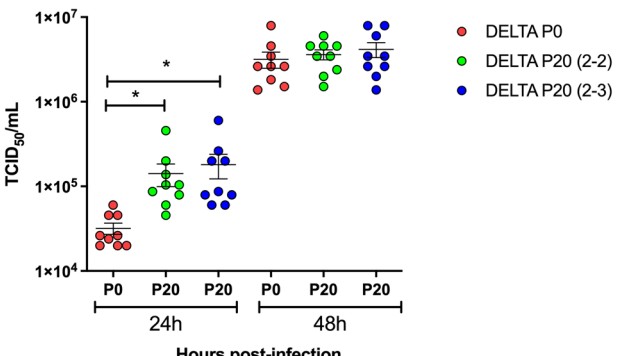

**Fig. 3 | Growth kinetics of P0 and P20 Delta viruses.** P0 and two independently evolved P20 (2-2 and 2-3) Delta viruses were used to infect Vero cells at a multiplicity of infection of 0.005. Cell-free supernatants were collected at 24 h and 48 h and used for viral titration. Results are expressed as mean ± SD $TCID_{50}$/mL. *$p = 0.02$ as determined using an unpaired, two-tailed $t$-tests assuming no difference between groups.

both lineages. The variant allele associated with Omicron was C22674T (Spike S371F) that arose de novo in 2/3 animals in P20 for Beta virus and 2/3 animals in P13 for Delta virus, persisting in later passages (Fig. 8, Supplementary Table 1). We thus observed an Omicron specific mutation in a study not involving Omicron viral lineages. The S371F mutation alters spike glycoprotein that interacts with human proteins GOLGA7 and ZDHHC5, which contribute to viral-cell entry[38] and palmitoylation that mediates viral infectivity[39], over-response to pathogens, and interferon production[40]. It has also been reported to undermine antibody response[41].

Of the variant alleles that changed in frequency, many involved viral proteins that interact with human proteins suggested to influence COVID-19 severity and susceptibility, representing druggable targets, and could explain phenotypes observed in this study (Supplemental Table 1)[42]. A929G (I222V), which arose and persisted in 1/3 animals infected with Delta in P17 (Supplemental Fig. 3), involves nsp2 via ORF1ab that interacts with GIGYF2[43]. C10341T and C10809T occur in nsp5 via ORF1ab (P3359L and P3515L), that interacts with HDAC2 that has been identified to enable immune evasion as an anti-immune effector[44]. C10341T arose in 1/3 animals in P13 for Delta and likely persisted for remaining passages, with its absence in P17 perhaps due to sequencing error (Supplemental Fig. 3). C10809T disappeared after P0 in Beta, reappearing in 1/3 animals in P17 (Supplemental Fig. 3). HDAC2 has four approved gene family-specific inhibitors, such as vorinostat, and one approved activator, theophylline[45]. G28237T disappeared after P0, then reappeared in 3/3 animals infected with Beta by P20 (Fig. 8). It occurred in ORF8 (R115L), which interacts with LOX, linked to COVID-19 severity and thrombosis[46], PLOD2, linked to COVID-19 and respiratory failure[42], and FKBP10, linked to poor COVID-19 prognoses[47]. *LOX* can be targeted by 3 approved medications, including the inducer cupric sulfate[45], *PLOD2* can be targeted by three approved medications, including a cofactor ascorbic acid[45], and FKBP10 can be stimulated by bleomycin[48].

Most alleles in the P0 Beta isolate, passaged several times in vitro in Vero cells, encoded a Spike protein with a tryptophan (W) at position 682 (C23606T), destroying the furin cleavage site (Table 1)[49]. We observed a rapid selection of viruses having the reference allele encoding an arginine (R)

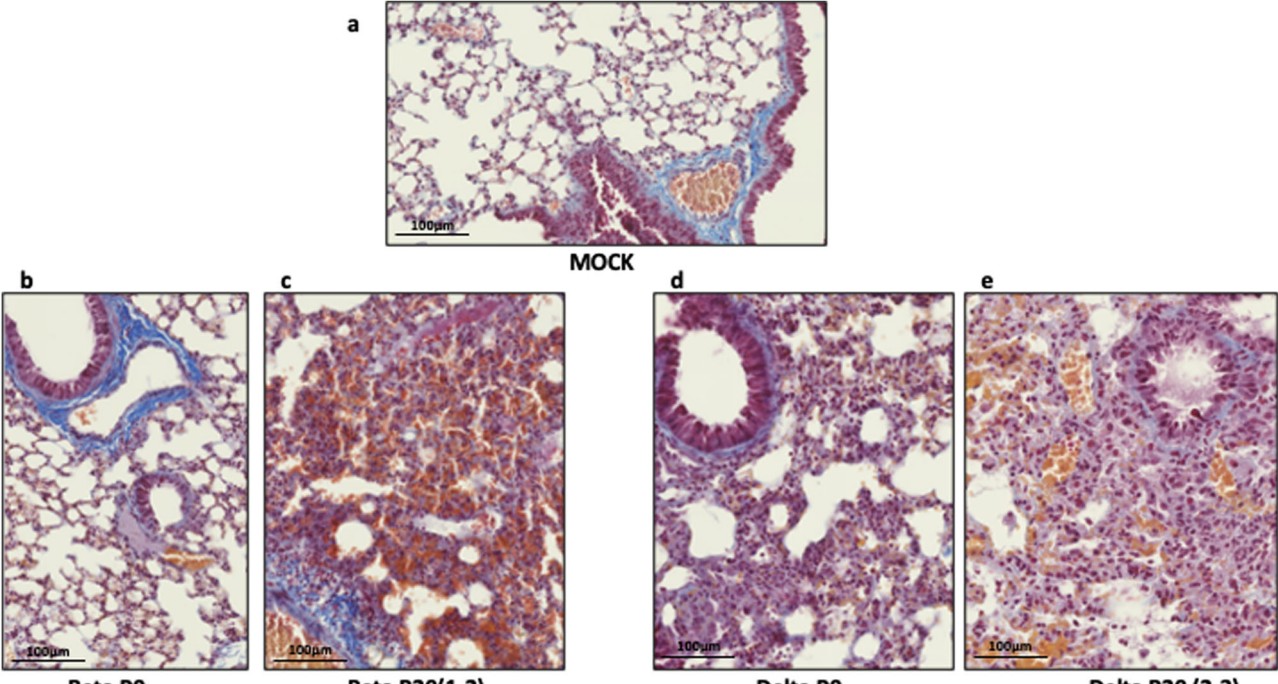

**Fig. 4 | Lung histology of mice infected with Beta and Delta P0 and P20 viruses.** Mice were mock-infected (**a**) or infected with 500 $TCID_{50}$ of Beta P0 (**b**), Beta P20(1-2) (**c**), Delta P0 (**d**) or (**e**) Delta P20(2-3) viruses. Lungs were harvested on day 6 (Beta) or day 7 (Delta) post infection, formalin fixed and processed for Carstairs staining. Scale bar represents 100 μm.

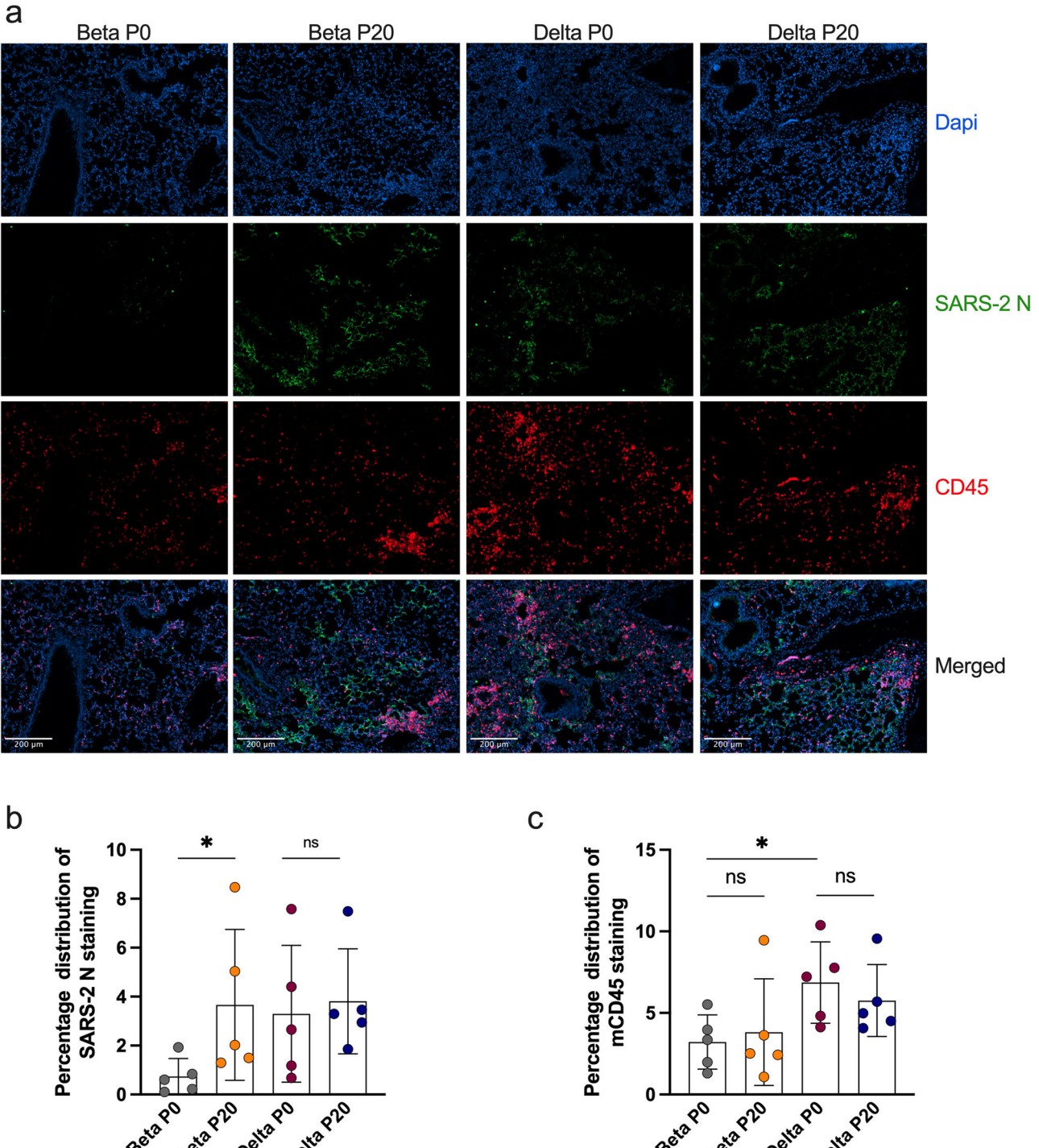

**Fig. 5 | Immunohistological staining of mouse lungs infected with Beta P0, Delta P0, Beta P20 (1-2) and Delta P20 (2-3) viruses. a** Infected lung tissues were stained with DAPI (blue/nuclei), anti-SARS CoV-2 nucleocapsid (N) protein (green), anti-mouse CD45 (red). Scale bar equals 200 μm. **b** Graphical (mean +/− SD) representation of SARS CoV-2 N (B) or CD45 **c** staining. Each dot represents data collected from one section of lung tissue from one mouse. Entire lung sections were imaged with representative images shown. *$p < 0.05$ and not significant (ns) determined using a two-tailed Mann-Whitney test, which assumes no difference between groups.

at position 682 and functional furin site in all animals infected with Beta variant starting in P10 (Table 1). This suggests that the furin site provides a growth advantage under in vivo conditions and supports a previous report indicating the Spike furin site being important for SARS-CoV-2 virulence in K18-ACE2 mice[49].

Nanopore focused sequencing of the S (spike) gene suggested several of these variant alleles were co-inherited by distinct Delta quasi-species. Spike mutations S371F and E1182G were observed to occur at identical Nanopore-derived allele frequencies in P17 for animal two and P20 for animal three, suggesting co-inheritance from P13 viruses. While only S371F and A892V were observed to have the same allele frequency in P13 with E1182G not detected by Nanopore in this passage, but detected by Illumina, this could be a product of sequencing error or changes, alongside observing co-inheritance of alleles at distinct times in distinct samples (Fig. 8). S371F

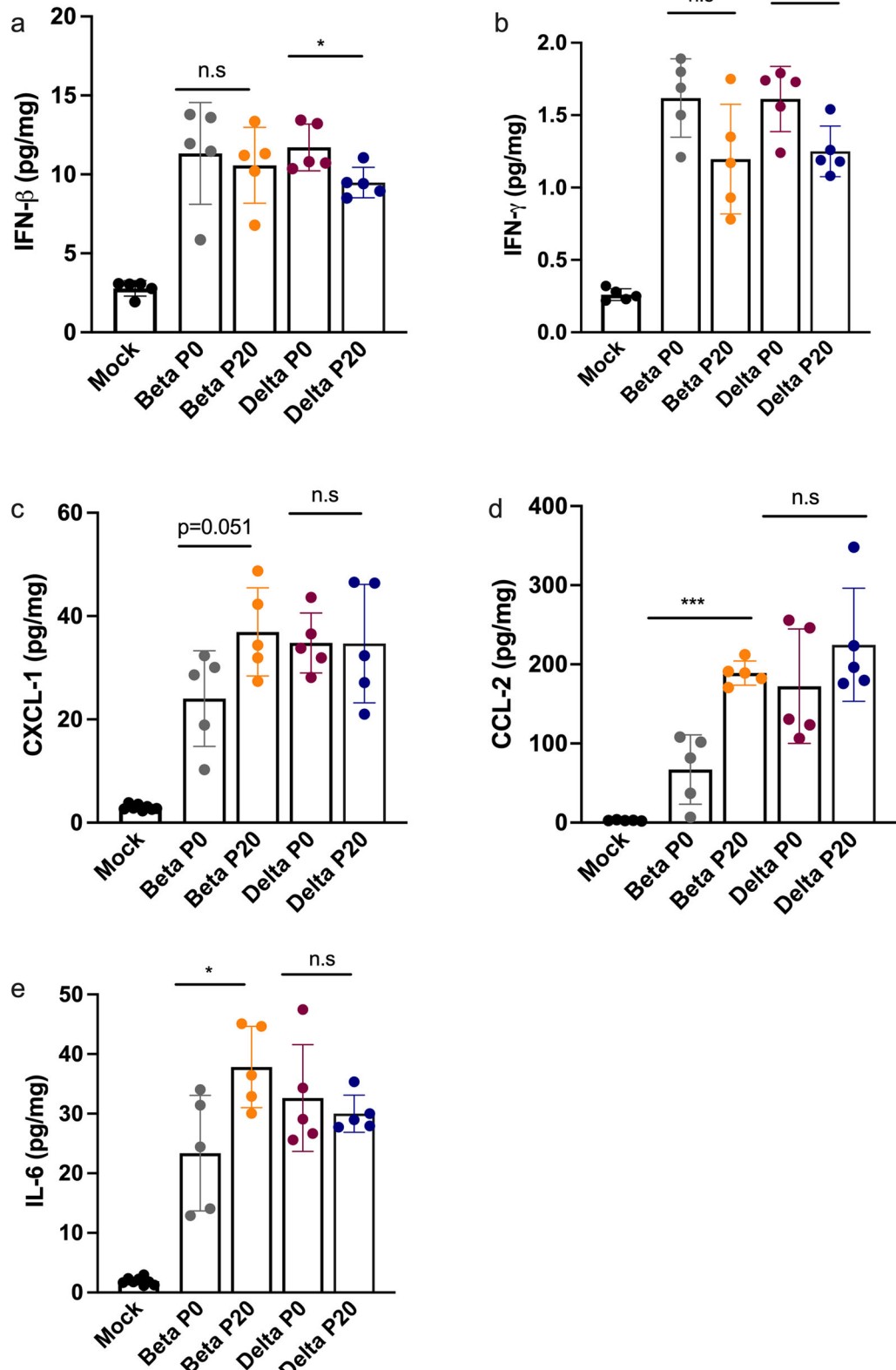

**Fig. 6 | Monitoring of cytokines in lungs of mice infected Beta P0, Delta P0, Beta P20 (1-2) and Delta P20 (2-3) viruses.** K18-ACE2 mice were mock-infected or infected with 500 TCID$_{50}$ of Beta P0, Beta P20 Delta P0 and Delta P20 viruses. On day 3 post-infection, mice were euthanized, lungs were harvested and homogenized. Concentrations of IFN-β (**a**), IFN-γ (**b**), CXCL-1 (**c**), CCL-2 (**d**), IL-6 (**e**) in lung homogenates were determines using the Luminex assay. *$p < 0.05$, ***$p < 0.005$, ns: not significant as determined using two-tailed unpaired $t$-test, comparing P0 and P20 groups, assuming no difference between groups.

**Fig. 7 | Neutralization of P0 and P20 viruses with sera from vaccinated subjects. a** Neutralization assay of wild-type (WT), Beta, and Delta viruses using sera from 24 vaccinated subjects. Individual results and mean ± SD neutralization titer against different SARS-CoV-2 isolates are presented. ****$p < 0.0001$ determined using one way-ANOVA. ns: not statistically significant. **b** Pair-wise comparison of the sera used in A in neutralization assay against P0 and P20 (1-2) Beta viruses. **c** Pair-wise comparison of the sera used in A in neutralization assay against P0 and P20(2-2) Delta viruses. ***$p < 0.002$ determined using a two-tailed, paired *t*-test, assuming no difference between groups.

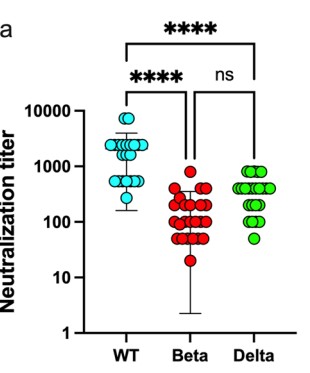
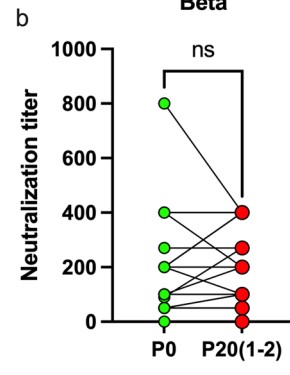
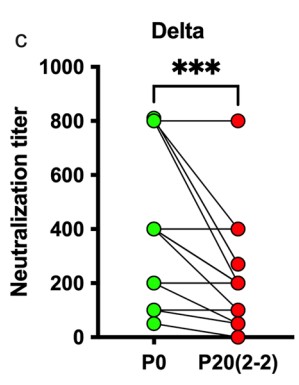

## Discussion

In this work, we show that more recent and human-adapted lineages of SARS-CoV-2, Beta and Delta, evolve in a setting of minimal selective pressures, Delta developed more clinically relevant changes than Beta with both yielding more pronounced lung disease and disease severity, and these phenotypic changes can be partially explained by discrete development and disappearance of alleles linked with key traits, such as antibody resistance and interferon suppression. Given our observance of these alleles with frequencies suggestive of minor quasi-species, it could suggest their significance in COVID-19, experimentally supporting others' conclusions in a controlled setting[36]. While the evolution of the original SARS-CoV-2 strain in Balb/c mice has been previously reported, with genetic changes used to explain phenotypic evolution[20,22], Beta and Delta lineages are more genetically adapted to humans with additional spike mutations, making a model more similar to humans with mice expressing human ACE2R more pressing. This also limits selection for murine Ace2 adaptation, establishing a baseline for a model to build off. We could not locate papers comparing the phenotypic changes of two lineages in a single study, offering controlled, experimental evidence for lineage-based evolutionary differences.

Antigenic variation developing in the absence of experimental selective pressures has been observed in other viruses, such as foot-and-mouth disease virus (FMDV), supporting the mechanism in SARS-CoV-2 and our observations[50,51]. Mechanistically, this could be mediated by biothermodynamics[52–56].

The Delta P0 virus, which lacked the N501Y mutation, never acquired it in our study. Acquisition of N501Y in humans was reported almost a year after SARS-CoV-2 epidemic's beginning, suggesting the mutation arose alongside growing herd immunity[57]. This could suggest that in the absence of selective immune pressure, such as neutralizing antibodies, or pressure to adapt to murine ACE2, N501Y is not favored (Fig. 8). Beta P0 harbored the N501Y mutation, which was previously reported to enable reference SARS-CoV-2's adaptation in non-transgenic mice[20]. This mutation remained unchanged in our study, suggesting that our virus could infect cells through the murine ACE2 receptor, which we demonstrated in non-transgenic C57BL6 mice using Beta P0 and P20 viruses (Supplementary Fig. 1). ACE2 transgene being more widely expressed than murine ACE2 could explain the greater SARS-CoV-2 RNA loads in K18 mouse lungs versus B6[58]. Beta's flexibility to use human and murine receptors could explain our isolate's greater virulence versus others, such as WT virus and Delta that lacked N501Y[58].

Evolved Beta and Delta modulated the inflammatory and antiviral responses differently than corresponding P0 viruses, both causing greater and more rapid weight loss and more severe lung damage without a significant change in detected inflammatory cells, suggesting mediation by overactivation of existing cells or systemic factors (Fig. 5). Beta P20 infection was associated with greater inflammatory cytokine production (CCL2,

CXCL1, IL-6) than P0 ($p < 0.005$, $p = 0.051$, $p < 0.05$, respectively) (Fig. 6). This occurred with the disappearance of several variant alleles, and the de novo development of S371F in P20 for 2/3 animals (Fig. 8, Supplemental Fig. 3). G23593T, or spike Q677H, which disappeared from Beta virus samples after P0, enhances treatment resistance and increase viral infectivity[59]. The reversion of Spike amino acid 677 to Q after passaging in mice could explain the increased neutralization activity for some of the sera against Beta P20 relative to P0 (Fig. 7), with non-receptor-binding-domain changes in spike believed to contribute to antibody escape[60]. Delta P20 infection yielded greater viral load than P0 with suppressed IFN-β and IFN-γ, suggesting acquisition of additional antiviral/immunomodulatory properties that could be mediated by S371F's (C22674T) effect on spike protein, which can suppress type I interferon expression[61]. S371F, which arose de novo and persisted in two animals in P20 for Beta and P13 for Delta virus (Fig. 8), is one of the 8 Omicron BA.2-specific spike mutations that induces a 27-fold reduction in the capacity of sotrovimab to neutralize BA.2, broadly affecting the binding of most RBD-directed antibodies (Table 1)[41]. A929G (ORF1ab I222V) also arose de novo in 1/3 animals, which could impact nsp2's suppression of GIGYF2's functions (Table 1, Supplementary Fig. 3)[43].

This study had limitations. We used K18-ACE2 transgenic mice but the mechanisms of new viral variants arising in humans could significantly differ, which is relevant as host factors are believed to contribute to viral variant allele diversity[62]. Our model lacked experimental selective pressures, while the COVID-19 pandemic has been accompanied by contact limitations and vaccines with the dominant SARS-CoV-2 evolutionary mechanism believed to be natural selection[14]. A baseline is required to effectively model these pressures. We studied Beta and Delta lineage virus, which are no longer the dominant lineages and our conclusions could vary with Omicron. This underscores the value in studying single-allele changes as they can capture fitness-related traits.

In conclusion, we demonstrate that more human-adapted SARS-CoV-2 lineages when passaged in mice expressing human ACE2 receptor evolve in the setting of minimal selective pressures, their changes vary by lineage, and accumulate clinically-relevant changes, such as antibody neutralization resistance.

## Methods

### Viruses
SARS-CoV-2 WT strain (LSPQ, B1 lineage) was obtained from the Laboratoire de Santé Publique du Québec ([LSPQ] Sainte-Anne-de-Bellevue, QC, Canada). SARS-CoV-2 Beta strain was obtained from BEI resources and SARS-CoV-2 Delta strain from the BC Centers for Disease Control. Unmodified SARS-CoV-2 strains were propagated on Vero cells (American Type Culture Collection, Manassas, Virginia, USA).

### Determination of the viral titer
Vero cells were plated in a 96 well plate ($2 \times 10^4$/well) and infected with 200 µl of serial dilution of the viral preparation or lung homogenate in the M199 media supplemented with 10 mM HEPES pH 7.2, 1 mM of sodium

**Table 1 | Annotations of select variants that changed in frequency across the study**

| Variant | Amino Acid Change | Annotation | Potential Interacting Human Protein | Human protein implicated with COVID-19 severity or susceptibility |
|---|---|---|---|---|
| A929G | ORF1ab I222V, nsp2 | Protein interacting* | GIGYF2, FKBP15*, WASHC4, RAP1GDS1, POR, eIF4E2*, SLC27A2 | GIGYF2[43] |
| C22674T | Spike S371F | Protein interacting, antibody escape, vaccine target, Omicron-associated variant allele[76] | GOLGA7, ZDHHC5 | GOLGA7, ZDHHC5[38] |
| A25107G | Spike E1182G | Protein interacting, vaccine target | GOLGA7, ZDHHC5 | GOLGA7, ZDHHC5[38] |
| G28237T | ORF8 R115L | Protein interacting* | COL6A1, PCSK6, LOX*, DNMT1*, NPC2, CISD3, ITGB1, PLAT, STC2, TOR1A, PLOD2*, INHBE, CHPF2, UGGT2, FBXL12, PLEKHF2, SMOC1, POFUT1, FKBP10*, ERLEC1, IL17RA, ADAMTS1, HS6ST2, SDF2, NEU1, GDF15, TM2D3, SIL1, EDEM3, ERP44, PVR, NGLY1, OS9, ADAM9, NPTX1, POGLUT2, POGLUT3, ERO1B, PLD3, FOXRED2, CHPF, PUSL1, HYOU1, MFGE8, FKBP7, GGH, EMC1 | COL6A1[79], LOX[46], PLAT[80], PLOD2[42], FBXL12[81], FKBP10[47], IL17RA[82], HS6ST2[83], NEU1[84], GDF15[85], ADAM9[86], PLD3[87] |

A * indicates a human protein that is druggable.

pyruvate, 2.5 g/L of glucose, 5 µg/mL *Plasmocin*® and 2% FBS. Three days post-infection plates were analyzed for cytophathic effet using a EVOS M5000 microscope (Thermo Fisher Scientific, Waltham, MA, USA) and titer determined using the Kärber method[63].

### Mouse models
B6.Cg-Tg(K18-hACE2)2Prlmn/J (stock#3034860) and B6 mice were purchased from the Jackson Laboratories (Bar Harbor, ME). All mouse studies were conducted in a BSL-3 laboratory. Mice of both sexes and between the ages of 7–9 weeks were used throughout the protocol. One K18-ACE2 mouse was intranasally infected with either B.1.351 (Beta) or B.1.617.2 (Delta) SARS-CoV-2 lineages. Following 3–4 days, the animals were sacrificed and lung homogenate collected. The homogenate was used to intranasally infect sequential animals, defined as passage. This was repeated for 10 passages. After passage 10 (P10), virus in lung homogenate was sequenced. K18-ACE2 mice were then intranasally infected with Beta P10 or Delta P10 with viruses passaged in three mice at a time for an additional 10 times (total of 20 passages) with virus in lung homogenate sequenced following passages 13, 17, and 20 (Fig. 1). Three viral stocks of Beta P20 (1-1, 1-2, 1-3) and three viral stocks of Delta P20 (2-1, 2-2, 2-3) viruses were then made from lung homogenates. K18-ACE2 mice were infected intranasally with 50050% tissue culture infectious dose ($TCID_{50}$) of P0 or P20 Beta (1-2) and Delta (2-3) viruses. Lung viral loads were determined on day 3 post-infection with weight loss and survival monitored daily for up to 9 days.

### Infectivity of P0 and P20 viruses to non-transgenic mice
C57BL6 mice were infected intranasally with 3000 $TCID_{50}$ of P0 and P20 Beta and Delta viruses. Three days later, mice were euthanized, and lungs collected for viral load (titer) determination. Lobes from the right lung were homogenated in PBS using the Omni Bead Ruptor Bead Mill homogenizer (Kennesaw, GA) and used for viral load determination.

### RNA extraction
Up to 30 mg of lung tissue were used for RNA extraction using the Bead Mill Tissue RNA Purification Kit and the Bead Mill Homogenizer (Kennesaw, GA). Tissue and 700 µl of lysis buffer were added to tubes containing 2.8 mm ceramic beads and kept on ice. Ten µl of anti-foaming agent were added before homogenization (4 m/s for 30 s). Samples were centrifuged and supernatants collected. An equal volume of 70% ethanol was added to each tube before RNA purification over Omni RNA Mini columns. RNA samples were eluted in 50 µl of water and stored frozen until used.

### Droplet digital PCR (ddPCR) quantitation of SARS CoV-2 RNA
SARS-CoV-2 viral RNA loads were determined using Droplet Digital PCR (ddPCR) supermix for probes without dUTP (Bio-Rad Laboratories Ltd.) and the QX200 Droplet Digital PCR System Workflow (Bio-Rad Laboratories Ltd.). The ddPCR primers and probes were previously reported[64].

### Multiplex cytokines quantification
Cytokines in mouse lung homogenates were measured using a custom ProcartaPlexTM Mouse Mix & Match Panels kit (Invitrogen, Waltham, MA, USA) on the Bio-Plex 200 (Bio-Rad Laboratories Ltd.).

### Histological analysis
For each group analyzed ($N = 5$ mice/group), the right lung lobe was extracted, fixed in formalin and paraffin embedded as described[29]. Lung sections were stained with Carstairs staining for histological analysis. Prior to conduct immunofluorescence assay, lung sections were deparaffined, hydrated then heat-induced epitope retrieval 16 h at 60 °C with Diva Decloaker solution (Biocare Medical, Pacheco, CA, USA). Immunostainings were preformed to detect SARS-CoV-2 N antigen and leukocyte infiltration using 20 µg/mL rabbit anti-N (Rockland chemicals, Limerick, PA, USA)/anti-rabbit IgG Alexa 488 (Jackson Immuno Research lab, West Grove, PA, USA) and 10 µg/mL biotinylated anti-CD45 antibodies (BD Bioscience, Franklin Lakes, NJ, USA)/anti Rat IgG Alexa Plus 647 (Thermo

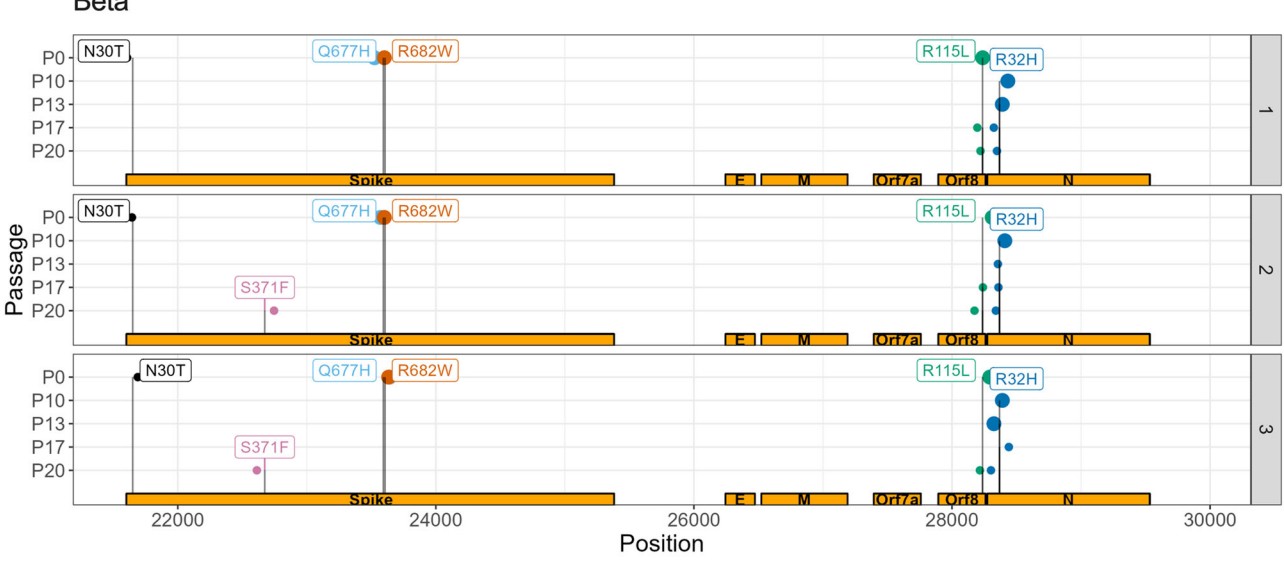

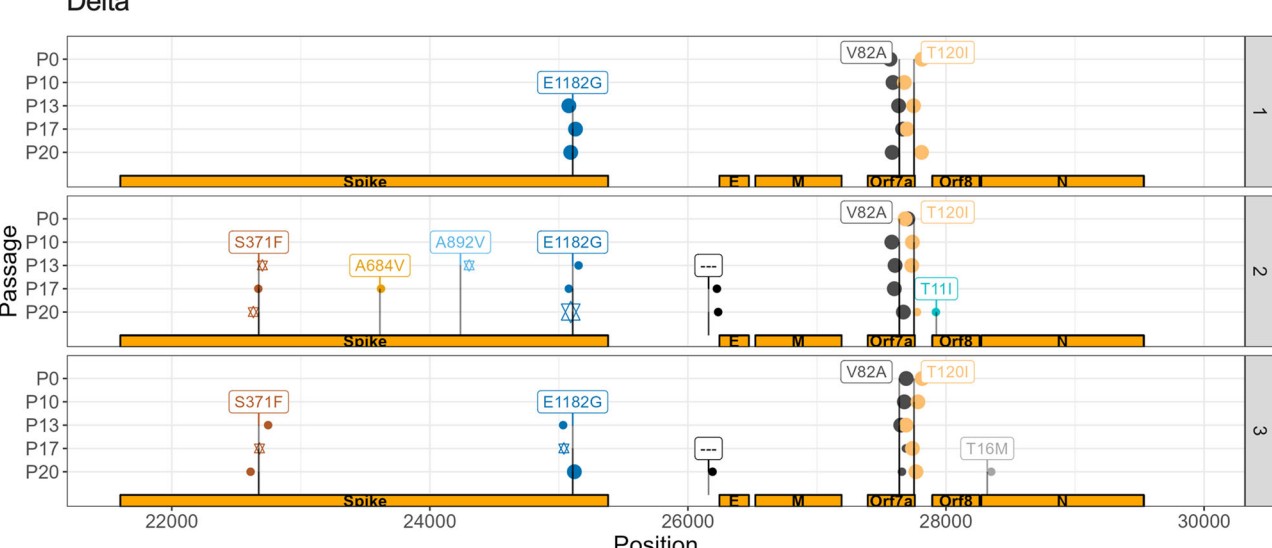

**Fig. 8 | More variant alleles of clinical relevance arose de novo and persisted in spike for Delta lineage compared to Beta, with key variants being co-inherited.** Each line and sample number refers to a select viral sample being passaged. P0 and P10 are identical across samples, given that it was passaged in a single animal up to P10. VAF: variant allele frequency. P refers to passage. Variant allele incidence by sample in Delta lineage virus. Larger points represent variant alleles with a within-sample, Illumina-derived allele-frequency of 1.0, smaller ones 0.5. Stars in a given passage represent alleles that are likely co-inherited per Nanopore sequencing.

Fisher Scientific, Waltham, MA, USA). Slide were imaged using Axioscan 7 instrument (Carl Zeiss Microscopy, New York, USA) then black and white adjustment were performed with Zen lite 3.7 (Carl Zeiss Microscopy). Quantification of positive area signal for N and CD45 staining were performed using Fiji (ImageJ) threshold analyse tools.

**Illumina viral sequencing**
RNA extracts were processed for reverse transcription (step a) and targeted SARS-CoV-2 amplification using the ARTIC V3 or V4.1 primer scheme (https://github.com/artic-network/primer-schemes/tree/master/nCoV-2019) (step b). Samples were purified (step b) and Nextera DNA Flex library preparation (step c) was performed for Illumina PE150 paired-end amplicon sequencing on a NovaSeq or MiSeq instrument at the McGill Genome Centre using best practices. Each sample was sequenced twice on different days to obtain a target of minimum 10 million reads per sample. The detailed protocols can be accessed with the following links.

Step a: https://doi.org/10.17504/protocols.io.bjgekjte.
Step b: https://doi.org/10.17504/protocols.io.ewov18e4ygr2/v2.

Step c: https://doi.org/10.17504/protocols.io.bjgnkjve.

**Nanopore viral sequencing**
RNA extracts were processed by reverse transcription with Lunascript[65]. Targeted SARS-COV-2 amplification was performed using five of the 29 Midnight primers[66]. We targeted the 4167 to 5359 bp region with the primer pairs SARSCoV_1200_5. To target the Spike region we used primers pairs SARSCoV_1200_23, SARSCoV_1200_25 SARSCoV_1200_22 and SARS-CoV_1200_24. Amplification of the primers was performed following Freed and Silander protocol[67]. Nanopore library preparation was made following Reiling et al. 2020 and libraries were sequenced on the PromethION 24 sequencer with PromethION Flow Cells V.9.4.1 for a total of 10 million reads[68].

**Genome data processing**
Following sequencing, the reads from each sample were used to call variants using Freebayes v1.3.6[69] and the results were saved as a VCF file which was used to compare genomic variation across passages. Others

have called variants with other tools such as DeepSNV[36] and VarScan[70]. DeepSNV was not used as it only investigates SNVs[71], when indels have been found to confer selective advantages[72]. Freebayes has been observed to perform slightly superiorly than VarScan when calling variants in wastewater samples[73]. The process is described briefly as follows: first, the raw reads from each sample were aligned with BWA MEM v0.7.17[74] to the reference strain genome sequence (NC_045512.2). The resulting alignments were sorted with duplicate reads flagged. The minimum number of aligned reads for each was logged, and each sample was again aligned and processed, then randomly down sampled to match this minimum number of reads. The minimum number of reads per sample in the first part of the study across passage 0 (P0) and P10 was 8,812,604. The minimum number of reads per sample in the second part of the study across P13, P17, and P20 was 34,253,784. All variants with a quality of <20 or a depth below 10 were removed from the analysis. Any variant alleles that overlapped with ARTIC sequencing primers were also removed. Merged VCF files, representing the combination of reads across two sequencing batches for each sample, were used for the paper's main results.

Nanopore reads were processed similarly using freebayes, although run a second time using freebayes with the haplotype-basis-alleles option to use statistical priors to further remove noise from reads. Given that Nanopore was used to validate Illumina calls and evaluate for co-inheritance, its raw output was not down sampled. Its raw output was aligned using minimap2 v2.24, then processed identically to Illumina reads.

### Variant annotation
Every variant allele was inputted into COVID-19 Ensembl variant effect predictor (VEP) (https://covid-19.ensembl.org/index.html) to determine functional consequences and the involved gene and amino acid change[75]. Variants whose VAF changed at least once across passages during the study were logged and input to the UCSC Genome Browser (https://genome.ucsc.edu/), checking for annotations of antibody escape, CD8 escape, vaccine targets, drug resistance, or variants of concern[76].

### Determination of spike allele co-inheritance
Alleles were determined to be co-inherited if the nanopore, statistically derived allele frequency was identical for multiple alleles in the same passage.

### Antibody neutralization studies
Sera from twenty-four healthy vaccinated (3 x) individuals were analyzed for neutralizing activity against WT, and Beta and Delta P0 and P20 viruses. Serially diluted sera (in quadruplicate) were incubated with 100 TCID$_{50}$ of virus for 1 h at 37˚C before addition to Vero E6 cells. Three days later, the 96-well plates were observed using an inverted microscope for signs of infection. Neutralization titers were determined and defined as the highest serum dilution preventing infection.

### Ethics
The study was conducted in accordance with the Declaration of Helsinki, and mouse protocols were approved by the Comité de protection des animaux de l'Université Laval. We have complied with all relevant ethical regulations for animal use. Repeated passaging of viruses into immune naïve mice does not preclude from potential adaptations resulting in viruses that are more virulent for mice. As a result, all samples containing serially-passaged virus were handled with the utmost precautions under BSL3 conditions until rendered not infectious using effective and proven inactivation methods. Sera obtained from consenting vaccinated subjects and in compliance with approval #2022-6204 delivered by the Centre Hospitalier Universitaire de Québec Ethics Review Board.

### Statistics and reproducibility
All statistical analyses were conducted using R/4.2.1. All statistical comparisons between passages were corrected for multiple testing using a Benjamini-Hochberg *p*-value adjustment[77]. Statistical tests included unpaired and paired t-tests and Mann-Whitney tests, as noted in figures. Viral loads and cytokine content were estimated using a minimum of five mice or samples per group. Neutralization assays were perfomed using sera from 24 vaccinated individuals. Mouse weight loss and survival were determined using 10 mice per group. Replicates are defined as distinct samples representing an identical time point or treatment dose.

### Reporting summary
Further information on research design is available in the Nature Portfolio Reporting Summary linked to this article.

### Data availability
The numerical source data behind the graphs in the Figs. 2–8 can be found in Supplementary Data 1. VCF files used to complete computational analyses are available at https://zenodo.org/records/10460318. Raw data file for Illumina and Nanopore sequencing can be found on the NCBI BioProject using accessing codes PRJNA1068670 and PRJNA1069731, respectively. All numerical source data for graphs/charts are provided in the supplementary material section.

### Code availability
The code to replicate analyses is available on *Zenodo*[78] *and at:* https://github.com/juliandwillett/SARS_CoV_2_Serial_Passaging_Study.

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

## Acknowledgements

This research was enabled in part by support provided by Calcul Québec (calculquebec.ca) and the Digital Research Alliance of Canada (alliance can .ca) #wst-164 (JR). This study was supported by the Canadian Institutes for Health Research (CIHR 476575) to JDSW, a CIHR operating grant to the Coronavirus Variants Rapid Response Network (CoVaRR-Net) (ARR-175622) to LF and JR, the Canadian foundation for innovation major strategic initiative (CFI_MSI#35444) (JR) and CIHR grant (GA1-117694) to LF. The funding body had no role in the design of the study or collection, analysis, interpretation, or reporting of the data. We thank Ines Colmegna for providing vaccinated patient sera. We thank Silvia Vidal for her guidance regarding SARS-CoV-2 evolutionary mechanisms. ChatGPT was used to decrease the word count in the abstract with output carefully examined for representing our results. We thank Dr Serge Rivest's imagery platform for the slide scanning service.

## Author contributions

J.D.S.W. conceived of the computational analyses needed for this study, completed all computational analysis of viral sequencing data, and wrote the majority of the manuscript. L.F. and J.R. conceived of the study. L.F. contributed to manuscript revisions, wrote sections relevant to animal models and antibody neutralization studies and performed analysis of results. J.R. supervised sequencing data generation and analysis. I.D. and L.G. performed all works involving live SARS-CoV-2 viruses. A.G. participated in the sequence analysis and methodological design of experiments. A.C.d. S.P.A. performed the Luminex assay and analyzed the data. É.L. performed the immunohistological analyses and analyzed the data. P.F. provided key reagents. J.H.G. contributed bioinformatics analysis review and manuscript revisions. J.-L.L., J.A.C., H.H.V.D., A.-M.R., S.L. and S.-H.C. supported the SARS-CoV-2 viral sequencing.

## Competing interests

All authors declare no competing interests.
