## [Peer Review File · Communications Biology]

Reviewers' comments:

Reviewer #1 (Remarks to the Author):

Overall, the manuscript is potentially interesting and flows well. However, the authors should consider the following information to elevate the clarity of their findings by following suggestions:

1) The Abstract is difficult to follow. The meaning of the first sentence is very difficult to follow: "The persistence of COVID-19 is partly due to viral evolution reducing vaccine and treatment efficacy". Also, this other sentence: "....., existing in different quasi-species at others". This sentence is also difficult to correlate with the Abstract: "Both are linked to mammalian GOLGA7 and ZDHHC5 interactions, which mediate viral-cell entry and antiviral response".

2) In the Introduction, line 48, the sentence: "Many RNA viruses, like SARS-CoV-2, exist as quasi-species meaning that within a given viral population, a multitude of alleles (major and minor) are present" could be re-phrased and completed as follows: "Many RNA viruses, like SARS-CoV-2, exist as quasispecies meaning that within a given viral population, a multitude of mutants are present subjected to a continuous process of genetic variation, competition, and selection". I would use the term "mutant" instead of "alleles". In addition to the paper of Sun et al. there are additional papers that could be also cited because they report replication of SARS-CoV-2 as viral quasispecies. For example:

Karamitros T, Papadopoulou G, Bousali M, Mexias A, Tsiodras S, Mentis A. 2020. SARS-CoV-2 exhibits intra-host genomic plasticity and low-frequency polymorphic quasispecies. *J Clin Virol* 131:104585.

Jary A, Leducq V, Malet I, Marot S, Klement-Frutos E, Teyssou E, Soulie C, Abdi B, Wirden M, Pourcher V, Caumes E, Calvez V, Burrel S, Marcelin AG, Boutolleau D. 2020. Evolution of viral quasispecies during SARS-CoV-2 infection. *Clin Microbiol Infect* 26:1560.e1–1560.e4.

Rueca M, Bartolini B, Gruber CEM, Piralla A, Baldanti F, Giombini E, Messina F, Marchioni L, Ippolito G, Di Caro A, Capobianchi MR. 2020. Compartmentalized replication of SARS-Cov-2 in upper vs. lower respiratory tract assessed by whole genome quasispecies analysis. *Microorganisms* 8: 1302.

Capobianchi MR, Rueca M, Messina F, Giombini E, Carletti F, Colavita F, Castilletti C, Lalle E, Bordi L, Vairo F, Nicastrì E, Ippolito G, Gruber CEM, Bartolini B. 2020. Molecular characterization of SARS-CoV-2 from the first case of COVID-19 in Italy. *Clin Microbiol Infect* 26:954–956.

Al Khatib HA, Benslimane FM, Elbashir IE, Coyle PV, Al Maslamani MA, Al-Khal A, Al Thani AA, Yassine HM. 2020. Within-host diversity of SARS-CoV-2 in COVID-19 patients with variable disease severities. *Front Cell Infect Microbiol* 10:575613.

Martínez-González B. et al. SARS-CoV-2 point mutation and deletion spectra and their association with different disease outcomes *Microb. Spectrum* 10(2): e0022122, 2022

Martínez-González B. et al. SARS-CoV-2 mutant spectra at different depth levels reveal an overwhelming abundance of low frequency mutations *Pathogens* 11(6):662, 2022

3) Line 66, ACE2 should be written with capital letters.

4) This is other cryptic sentence very difficult to understand (lines 80-81): Understanding longitudinal viral changes in a mammalian host is pressing.

5) In the first sentence of the Results, authors explained that mice were infected with early and late passaged viruses. Where have been these viruses described? Is there any previously manuscript published?

6) In Figure 2, why the lines for Beta viruses finish at day 6 and 9? If the mice have already died at those days, please indicate. The statistical significance (the asterisks) in Figure 2B is missing. Also, to complete the figure, it would be interesting to calculate the statistical significance of Figure 2C and D considering the entire line of the two variables (p0 vs p20) with a t-test.

7) Why in Figure 3 authors have now two different Delta P20 viruses? What lineage do they use to perform the experiments described in Figure 2?

8) Images in Figure 4 are very small to appreciate the details.

9) In this sentence: "On day 3 post infection, Beta P20 virus induced greater production of proinflammatory CXCL1, CCL2 and IL-6 versus P0 (Figures 6A-C)", the reference to the Figure is incorrect, I think it should refer to Figure 6C,D,E. Also, apply to the panels where IFNs were measured.

10) I am again confused. Regarding Supplementary Figure 1, the authors says: repeated passages in K18-ACE2.....are the viral passages performed in cell culture or in mice??? And also, why do they present now tree lineages of Betain panel B? There is a total lack of symmetry throughout the paper.

11) Sorry, but I do not understand Supplemental Figure 2. What does VAT and MAF mean? How do you calculate the mutant frequency?

12) There are previous examples on antigenic variation in the absence of selective pressures for other viruses such foot-and-mouth disease virus that can be cited:

Diez et al. J. Gen. Virol. 70, 3281, 1989

Domingo et al J. Gen. Virol. 74: 2039, 1993

Reviewer #2 (Remarks to the Author):

RE: Willett et al. "SARS-CoV-2 evolution in the absence of selective immune pressures, results in antibody resistance, interferon suppression and phenotypic differences by lineage."

SUMMARY:

Willett et al. demonstrated that serial passages of SARS-CoV-2 in mice expressing human ACE2 receptors in a minimal to moderate selective pressure result in human-adapted SARS-CoV-2 lineages with the accumulation of genetic changes relevant to antibody resistance.

MAJOR COMMENTS

This is a relatively complete genetic analysis of SAR-CoV-2 evolution in vivo in mouse models, with potentially important clinical implications. The study was well-designed and implemented. Some changes should be made prior to its acceptance for publication.

Concerning the selective pressure in this study, the authors believe that their work "did not constitute gain-of-function research as virus was unmodified and represented circulating variants with the study conducted without experimental selective pressures (Page 16)." This and other similar statements in the main text should be interpreted cautiously. The reviewer considers that serial passages of viruses actually constitute selective pressures for viruses that are potentially having gain-of-function characteristics. The authors need to revise their statements in the title and the main text. Also, the biosafety level (BSL-3) should be elevated to higher levels for processing all viruses-containing samples.

Table 1: There are many interesting and potentially important human interaction proteins (e.g., ZDHHC5 and HGOLGA7). However, these data are descriptive in nature without experimental evidence that supports their relevance in SAR-CoV-2 infectivity in this study. The authors may consider providing additional experiments or providing this table in Supplementary Information.

OTHER / MINOR COMMENTS:

Line 26: Could you specify Wuhan-like SARS-CoV-2? Which strain do you refer to? If possible, avoid using "Wuhan" throughout the text.

Line 39: Treatment resistance is not standard terminology. Consider changing to "therapeutic Resistance."

Line 98: "... study used transgenic mice expressing the human ACE2 receptor." Please provide more descriptions about the transgenic expression ACE2 receptor in mouse lungs and other tissues.

Line 108: "... more infectious virus after 24h of infection versus Delta P0 virus ($p = 0.02$) (Figure 3)." Please change "versus" to "than."

Line 124: need to briefly explain, to a broader readership, the K18-ACE mice and the rationale to use this model in the study.

Line 135-136: change "vaccinated subjects" to "vaccinated human subjects"

Line 154: Give the full term for abbreviation(s) at its first appearance in the main text (e.g., VAF on Page 7 and TCID50 within line 289).

Figure 4: missing labels A, B, C, and D in this figure.

Reference citations: In many cases, more citations are needed to reinforce the findings and statements. Please provide additional references if available.

Line 384: Provide a reference for the Benjamini-Hochberg p-value adjustment.

Reviewer #3 (Remarks to the Author):

Review for the manuscript "SARS-CoV-2 evolution in the absence of selective immune pressures, results in antibody 2 resistance, interferon suppression and phenotypic differences by lineage" (COMMSBIO-23-1680-T, Communications Biology)

July 8, 2023

SARS-CoV-2 is the best studied virus from the perspectives of molecular biology, epidemiology, immunology, chemistry and biothermodynamics. The manuscript "SARS-CoV-2 evolution in the absence of selective immune pressures, results in antibody 2 resistance, interferon suppression and phenotypic differences by lineage" represents a significant step in the research of time evolution of SARS-CoV-2. Except for the contribution to the research on viruses, it also represents a contribution to the research on viral evolution.

The manuscript analyzes especially well the evolution of SARS-CoV-2 from the perspectives of epidemiology, biology, immunology and molecular biology. However, it is possible to include into the analysis the perspective of biothermodynamics, which gives a mechanistic model for changes in infectivity and pathogenicity during evolution of SARS-CoV-2. This research is available in the literature

Popovic, M., Martin, J. H., & Head, R. J. (2023). COVID infection in 4 steps: Thermodynamic considerations reveal how viral mucosal diffusion, target receptor affinity and furin cleavage act in concert to drive the nature and degree of infection in human COVID-19 disease. *Heliyon*, 9(6), e17174. <https://doi.org/10.1016/j.heliyon.2023.e17174>

Popovic, M. (2023). SARS-CoV-2 strain wars continues: Chemical and thermodynamic characterization of live matter and biosynthesis of Omicron BN.1, CH.1.1 and XBC variants. *Microbial Risk Analysis*, 24, 100260. <https://doi.org/10.1016/j.mran.2023.100260>

Popovic, M., Pantović Pavlović, M., & Pavlović, M. (2023). Ghosts of the past: Elemental composition, biosynthesis reactions and thermodynamic properties of Zeta P.2, Eta B.1.525, Theta P.3, Kappa B.1.617.1, Iota B.1.526, Lambda C.37 and Mu B.1.621 variants of SARS-CoV-2. *Microbial risk analysis*, 24, 100263. <https://doi.org/10.1016/j.mran.2023.100263>

Popovic, M., & Popovic, M. (2022). Strain Wars: Competitive interactions between SARS-CoV-2 strains are explained by Gibbs energy of antigen-receptor binding. *Microbial risk analysis*, 21, 100202. <https://doi.org/10.1016/j.mran.2022.100202>

Popovic, M. (2022). Biothermodynamics of Viruses from Absolute Zero (1950) to Virothermodynamics (2022). *Vaccines*, 10(12), 2112. MDPI AG. Retrieved from <http://dx.doi.org/10.3390/vaccines10122112>

The mechanistic model and driving force for changes in infectivity and pathogenicity can give a response to the question of why during evolution there has been an increase in infectivity and maintenance or slight change in pathogenicity of the new variants of SARS-CoV-2, as well as suppression of earlier variants by the new variants.

In summary, the manuscript provides a significant contribution to the field. This reviewer believes that it deserves high priority for publication after a minor revision.

Reviewer 1

1. The Abstract is difficult to follow. The meaning of the first sentence is very difficult to follow: “The persistence of COVID-19 is partly due to viral evolution reducing vaccine and treatment efficacy”. Also, this other sentence: “....., existing in different quasi-species at others”. This sentence is also difficult to correlate with the Abstract: “Both are linked to mammalian GOLGA7 and ZDHHC5 interactions, which mediate viral-cell entry and antiviral response”.

Thank you for bringing this to our attention. We have edited these sentences to make them easier to follow, which are shown below.

Lines 25-26: The **persistence of SARS-CoV-2 despite the development of vaccines and a degree of herd immunity** is partly due to viral evolution reducing vaccine and treatment efficacy.

Lines 34-37: S371F, an Omicron-characteristic mutation, was co-inherited at times with spike E1182G per Nanopore sequencing, existing in different **within-sample viral variants** at others. Both **S371F and E1182G** are linked to mammalian GOLGA7 and ZDHHC5 interactions, which mediate viral-cell entry and antiviral response.

2. In the Introduction, line 48, the sentence: “Many RNA viruses, like SARS-CoV-2, exist as quasi-species meaning that within a given viral population, a multitude of alleles (major and minor) are present” could be re-phrased and completed as follows: “Many RNA viruses, like SARS-CoV-2, exist as quasispecies meaning that within a given viral population, a multitude of mutants are present subjected to a continuous process of genetic variation, competition, and selection”. I would use the term “mutant” instead of “alleles”. In addition to the paper of Sun et al. there are additional papers that could be also cited because they report replication of SARS-CoV-2 as viral quasispecies. For example:

Karamitros T, Papadopoulou G, Bousali M, Mexias A, Tsiodras S, Mentis A. 2020. SARS-CoV-2 exhibits intra-host genomic plasticity and low-frequency polymorphic quasispecies. *J Clin Virol* 131:104585.

Jary A, Leducq V, Malet I, Marot S, Klement-Frutos E, Teyssou E, Soulie C, Abdi B, Wirden M, Pourcher V, Caumes E, Calvez V, Burrel S, Marcelin AG, Boutolleau D. 2020. Evolution of viral quasispecies during SARS-CoV-2 infection. *Clin Microbiol Infect* 26:1560.e1–1560.e4.

Rueca M, Bartolini B, Gruber CEM, Piralla A, Baldanti F, Giombini E, Messina F, Marchioni L, Ippolito G, Di Caro A, Capobianchi MR. 2020.

Compartmentalized replication of SARS-Cov-2 in upper vs. lower respiratory tract assessed by whole genome quasispecies analysis. *Microorganisms* 8: 1302.

Capobianchi MR, Rueca M, Messina F, Giombini E, Carletti F, Colavita F, Castilletti C, Lalle E, Bordi L, Vairo F, Nicastrì E, Ippolito G, Gruber CEM, Bartolini B. 2020. Molecular characterization of SARS-CoV-2 from the first case of COVID-19 in Italy. *Clin Microbiol Infect* 26:954–956.

Al Khatib HA, Benslimane FM, Elbashir IE, Coyle PV, Al Maslamani MA, Al-

Khal A, Al Thani AA, Yassine HM. 2020. Within-host diversity of SARS-CoV-2 in COVID-19 patients with variable disease severities. *Front Cell Infect Microbiol* 10:575613.

Martínez-González B. et al. SARS-CoV-2 point mutation and deletion spectra and their association with different disease outcomes *Microb. Spectrum* 10(2): e0022122, 2022

Martínez-González B. et al. SARS-CoV-2 mutant spectra at different depth levels reveal an overwhelming abundance of low frequency mutations *Pathogens* 11(6):662, 2022

Thank you for these suggestions. We have revised the introduction following your suggestion, as shown below. We have also included citations for these manuscripts, which are shown below and have been added to our reference list.

Lines 47-49: Many RNA viruses, like SARS-CoV-2, exist as quasispecies, meaning viral populations contain a multitude of mutants subjected to continuous selection, competition, and genetic variation³⁻⁹.

3. Line 66, ACE2 should be written with capital letters.

Thank you for picking this up. Mouse genes are typically written in lowercase, in comparison to human gene names, so we did not make this correction.

4. This is other cryptic sentence very difficult to understand (lines 80-81):
Understanding longitudinal viral changes in a mammalian host is pressing.

Thank you for highlighting this point of improvement. We have made this sentence easier to read, as is shown below.

Lines 81-82: Understanding how SARS-CoV-2 evolves as it passes between mammalian hosts is pressing.

5. In the first sentence of the Results, authors explained that mice were infected with early and late passaged viruses. Where have been these viruses described? Is there any previously manuscript published?

Thank you for highlighting this point of clarification. These early and late-passaged viruses correspond to virus prior to passaging them in our mice (early) and virus following twenty passages in mice. This method was adapted by works by Gu et al. (references 20 and 22). We have edited this first sentence to clarify this, as shown below.

Lines 98-100: To study evolution on the organism level, we compared clinical scores of mice infected with virus before we passaged it in mice (passage [P] 0, or early passaged virus) and after twenty passages (P20, or late-passaged virus), following a protocol adapted from the work of Gu et al²⁰.

6. In Figure 2, why the lines for Beta viruses finish at day 6 and 9? If the mice have already died at those days, please indicate. The statistical significance (the asterisks) in Figure 2B is missing. Also, to complete the figure, it would be interesting to calculate the statistical significance of Figure 2C and D considering the entire line of the two variables (p0 vs p20) with a t-test.

Thank you for identifying these points of clarification. The figure was modified with corresponding statistics added. The lineage of the P20 used was also specified to make it clearer. We also modified the figure to include the survival data in relation to the weight loss. Statistics on weight loss were also added, as suggested.

7. Why in Figure 3 authors have now two different Delta P20 viruses? What lineage do they use to perform the experiments described in Figure 2?

Thank you for identifying this point. We have added clarifications on this issue and modified figure 2 and 3 to indicate the viral isolates used.

Lines 295-298:). **Three viral stocks of Beta P20 (1-1, 1-2, 1-3) and three viral stocks of Delta P20 (2-1, 2-2, 2-3) viruses** were then made from lung homogenates. K18-ACE2 mice were infected intranasally with 500 **50% tissue culture infectious dose (TCID₅₀)** of P0 or P20 **Beta (1-2) and Delta (2-3) viruses**.

8. Images in Figure 4 are very small to appreciate the details.

Thank you for identifying this. We have now modified figure 4 to include the zoomed-in images only.

9. In this sentence: “On day 3 post infection, Beta P20 virus induced greater production of proinflammatory CXCL1, CCL2 and IL-6 versus P0 (Figures 6A-C)”, the reference to the Figure is incorrect, I think it should refer to Figure 6C,D,E. Also, apply to the panels where IFNs were measured.

Thank you for identifying this. We have corrected these labels for both Beta and Delta relevant changes.

10. I am again confused. Regarding Supplementary Figure 1, the authors says: repeated passages in K18-ACE2.....are the viral passages performed in cell culture or in mice??? And also, why do they present now tree lineages of Beta in panel B? There is a total lack of symmetry throughout the paper.

Thank you for identifying this point. We have clarified that non-transgenic mice were infected with virus isolated from passaging in K18-ACE2 mice, as shown below. As in other figures in the paper, except for Figure 3, we included a comparison of viral trait metrics by lineage (Beta and Delta) and passage given our research question of how these variables influenced viral traits. We did not include Beta in Figure 3 because it did not significantly change.

Lines 132-134: We next assessed whether repeated passages in K18-ACE2 mice would select for viruses capable of infecting non transgenic mice. C57BL6 (B6) mice were infected intranasally with P0 and P20 Beta and Delta viruses **isolated from serially-infected K18-ACE2 mice**

11. Sorry, but I do not understand Supplemental Figure 2. What does VAT and MAF mean? How do you calculate the mutant frequency?

Thank you for identifying this point of improvement. We have revised the figure legend to make it clearer. The mutant frequency was calculated using the variant calling software, BWA MEM or Freebayes, as detailed in the Methods (*Genome data processing*).

Lines 527-528: Each N refers to the number of variant alleles for each **variant allele frequency (VAF), determined using freebayes**, and in total against the NC_045512.2 reference sequence

12. There are previous examples on antigenic variation in the absence of selective pressures for other viruses such foot-and-mouth disease virus that can be cited:

Diez et al. J. Gen. Virol. 70, 3281, 1989
Domingo et al J. Gen. Virol. 74: 2039, 1993

Thank you for providing these helpful references. We have commented on these papers in the discussion, as shown below.

Lines 227-230: **Antigenic variation developing in the absence of experimental selective pressures has been observed in other viruses, such as foot-and-mouth disease virus (FMDV), supporting the mechanism in SARS-CoV-2 and our observations^{46,47}. Mechanistically, this could be mediated by biothermodynamics⁵²⁻⁵⁶.**

Reviewer 2

1. Concerning the selective pressure in this study, the authors believe that their work “did not constitute gain-of-function research as virus was unmodified and represented circulating variants with the study conducted without experimental selective pressures (Page 16).” This and other similar statements in the main text should be interpreted cautiously. The reviewer considers that serial passages of viruses actually constitute selective pressures for viruses that are potentially having gain-of-function characteristics. The authors need to revise their statements in the title and the main text. Also, the biosafety level (BSL-3) should be elevated to higher levels for processing all viruses-containing samples.

Thank you for ensuring we clearly communicate how we ensured safety for the public in our work. This conclusion was reached through close collaboration with our institutional biosafety board. We will continue close collaboration with our biosafety board for future experiments.

The title was modified to: **Short-term SARS-CoV-2 passaging in immune-naïve mice** results in **virus acquiring** antibody resistance, interferon suppression and phenotypic differences by lineage.

Lines 392-395: **Repeated passaging of viruses into immune naïve mice does not preclude from potential adaptations resulting in viruses that are more virulent for mice. As a result, all samples containing serially-passaged virus were handled with the utmost precautions under BSL3 conditions until rendered not infectious using effective and proven inactivation methods.**

2. Table 1: There are many interesting and potentially important human interaction proteins (e.g., ZDHHC5 and HGOLGA7). However, these data are descriptive in nature without experimental evidence that supports their relevance in SAR-CoV-2 infectivity in this study. The authors may consider providing additional experiments or providing this table in Supplementary Information.

Thank you for ensuring we best present our findings. We hope to keep this aspect of the table in the main body of the paper as it highlights the close relationship between viral and host genetics.

3. Line 26: Could you specify Wuhan-like SARS-CoV-2? Which strain do you refer to? If possible, avoid using “Wuhan” throughout the text.

Thank you for this suggestion. We have revised this sentence and all other instances of “Wuhan” in the manuscript, as shown below. The founder strain was B1 lineage, as mentioned in the methods on line 272.

Lines 26-28: Serial infections of **wild-type (WT)** SARS-CoV-2 in Balb/c mice yielded mouse-adapted strains with greater infectivity and mortality.

Line 144: being greatest against the original **WT** strain and lowest for the Beta isolate

Line 242: isolate's greater virulence versus others, such as **WT virus** and Delta that lacked N501Y

Line 281: **Viruses**. SARS-CoV-2 **WT** strain (LSPQ, B1 lineage) was obtained from

Line 385: were analyzed for neutralizing activity against **WT**, and Beta and Delta P0 and P20 viruses

Line 494: Neutralization assay of **WT**, Beta, and Delta viruses using sera from 24 vaccinated subjects

4. Line 39: Treatment resistance is not standard terminology. Consider changing to "therapeutic Resistance."

Thank you for your suggestion. We have made this revision.

5. Line 98: "... study used transgenic mice expressing the human ACE2 receptor." Please provide more descriptions about the transgenic expression ACE2 receptor in mouse lungs and other tissues.

Thank you for this suggestion. We have added this description to the manuscript, which comes from the Jackson laboratory website, which we mention in the Methods.

Line 102: expressing the human ACE2 receptor **in airway, liver, kidney, and gastrointestinal epithelium**

6. Line 108: "... more infectious virus after 24h of infection versus Delta P0 virus (p = 0.02) (Figure 3)." Please change "versus" to "than."

Thank you. We made this change.

7. Line 124: need to briefly explain, to a broader readership, the K18-ACE mice and the rationale to use this model in the study.

Thank you. We have added this rationale to the manuscript, as shown below.

Lines 99-102: Unlike Gu *et al.*'s study that required adaptation of SARS-CoV-2 to a murine Ace2 receptor for infectivity, our study used transgenic mice expressing the human ACE2 receptor **in airway, liver, kidney, and gastrointestinal epithelium, making the results more relevant for humans²⁰**.

8. Line 135-136: change "vaccinated subjects" to "vaccinated human subjects"

Thank you. We made this change.

9. Line 154: Give the full term for abbreviation(s) at its first appearance in the main text (e.g., VAF on Page 7 and TCID50 within line 289).

Thank you. We made these revisions.

10. Figure 4: missing labels A, B, C, and D in this figure.

Thank you. We made these revisions.

11. Reference citations: In many cases, more citations are needed to reinforce the findings and statements. Please provide additional references if available.

Thank you for this suggestion. We went through the manuscript and added further references, in addition to the additional references suggested by the other reviewers. These references included papers that also studied COVID-19 in K18-hACE2 mice, although none had studied it in the setting of evolution (Zheng et al 2021, Lee et al 2022, Tarres-Freixas et al 2022).

12. Line 384: Provide a reference for the Benjamini-Hochberg p-value adjustment.

We have added this reference (Benjamini and Hochberg 1995).

Reviewer 3

1. It is possible to include into the analysis the perspective of biothermodynamics, which gives a mechanistic model for changes in infectivity and pathogenicity during evolution of SARS-CoV-2. This research is available in the literature

Popovic, M., Martin, J. H., & Head, R. J. (2023). COVID infection in 4 steps: Thermodynamic considerations reveal how viral mucosal diffusion, target receptor affinity and furin cleavage act in concert to drive the nature and degree of infection in human COVID-19 disease. *Heliyon*, 9(6), e17174. <https://doi.org/10.1016/j.heliyon.2023.e17174>

Popovic, M. (2023). SARS-CoV-2 strain wars continues: Chemical and thermodynamic characterization of live matter and biosynthesis of Omicron BN.1, CH.1.1 and XBC variants. *Microbial Risk Analysis*, 24, 100260. <https://doi.org/10.1016/j.mran.2023.100260>

Popovic, M., Pantović Pavlović, M., & Pavlović, M. (2023). Ghosts of the past: Elemental composition, biosynthesis reactions and thermodynamic properties of Zeta P.2, Eta B.1.525, Theta P.3, Kappa B.1.617.1, Iota B.1.526, Lambda C.37 and Mu B.1.621 variants of SARS-CoV-2. *Microbial risk analysis*, 24,

100263. <https://doi.org/10.1016/j.mran.2023.100263>

Popovic, M., & Popovic, M. (2022). Strain Wars: Competitive interactions between SARS-CoV-2 strains are explained by Gibbs energy of antigen-receptor binding. *Microbial risk analysis*, 21, 100202. <https://doi.org/10.1016/j.mran.2022.100202>

Popovic, M. (2022). Biothermodynamics of Viruses from Absolute Zero (1950) to Virothermodynamics (2022). *Vaccines*, 10(12), 2112. MDPI AG. Retrieved from <http://dx.doi.org/10.3390/vaccines10122112>

Thank you for these reference suggestions. We have added them our text, as shown below.

Lines 225-226: **Mechanistically, this could be mediated by biothermodynamics⁵²⁻⁵⁶.**

REVIEWERS' COMMENTS:

Reviewer #1 (Remarks to the Author):

I would like to accept the manuscript with the new corrections made by the authors.

Reviewer #2 (Remarks to the Author):

The reviewer found few typo errors in online submission, which may be resulted in cut and paste of the word document with line numbers: For example: number "1" and "2" in title: "Short-term SARS-CoV-2 passaging in immune-naïve mice 1 results in virus acquiring 2 antibody resistance, interferon suppression and phenotypic differences by lineage." Please check it globally.

"Table 1. Annotations of select variants that changed in frequency across the study" presented in the revision is the truncated version of the original Table 1. It is much less informative than the original one. If the authors would like to present Table 1 in the main article (rather than supplementary information), use the original one noting that they are "potential interacting human proteins."

Reviewer #3 (Remarks to the Author):

The authors did a great job on the revision. The manuscript is suitable for publication in its current form.

Response to reviewer #2

Reviewer #2:

Remarks to the Author:

The reviewer found few typo errors in online submission, which may be resulted in cut and paste of the word document with line numbers: For example: number "1" and "2" in title: "Short-term SARS-CoV-2 passaging in immune-naïve mice 1 results in virus acquiring 2 antibody resistance, interferon suppression and phenotypic differences by lineage." Please check it globally.

"Table 1. Annotations of select variants that changed in frequency across the study" presented in the revision is the truncated version of the original Table 1. It is much less informative than the original one. If the authors would like to present Table 1 in the main article (rather than supplementary information), use the original one noting that they are "potential interacting human proteins."

Response

Thank you for these recommendations. We have revised the entire document. We have reverted Table 1 to the prior iteration, noting the first protein column as "Potentially Interacting Human Proteins."